# The Assessment of the Dynamics of Changes in the Torques of Redressing and Derotating Forces Acting on the Spine During Active Kyphosis-Deepening Exercises in the Dynamic Individual Stimulation and Control for Spine Device (DISC4SPINE, D4S)

**DOI:** 10.3390/jcm13247746

**Published:** 2024-12-18

**Authors:** Tomasz Szurmik, Karol Bibrowicz, Patrycja Romaniszyn-Kania, Damian Kania, Katarzyna Ogrodzka-Ciechanowicz, Piotr Kurzeja, Andrzej W. Mitas

**Affiliations:** 1Faculty of Arts and Educational Science, University of Silesia, 41-400 Cieszyn, Poland; tomasz.szurmik@us.edu.pl; 2Cavalry Captain Witold Pilecki State University of Małopolska in Oświęcim, 32-600 Oświęcim, Poland; 3Science and Research Center of Body Posture, Kazimiera Milanowska College of Education and Therapy, 61-473 Poznan, Poland; bibrowicz@wp.pl; 4Faculty of Biomedical Engineering, Silesian University of Technology, 44-100 Zabrze, Poland; patrycja.romaniszyn-kania@polsl.pl (P.R.-K.); andrzej.mitas@polsl.pl (A.W.M.); 5Department of Kinesiotherapy, Faculty of Physiotherapy, The Jerzy Kukuczka University of Physical Education in Katowice, 40-065 Katowice, Poland; d.kania@awf.katowice.pl; 6Institute of Clinical Rehabilitation, Faculty of Motor Rehabilitation, University of Physical Education, 31-571 Krakow, Poland; 7Institute of Health Sciences, Academy of Applied Sciences, 34-400 Nowy Targ, Poland; piotrkurzeja@op.pl

**Keywords:** diagnostics, human body, physiotherapy, scoliosis, forces

## Abstract

**Background**: The study aimed to assess the dynamics of changes in the torques of derotating and redressing forces acting on the apexes of deformation curvature arches during active, kyphosis-inducing exercises using the D4S device. **Methods**: The study group included 12 girls aged 9 to 10 years (age X = 9.36, SD = 1.52; weight X = 31.34 kg, SD = 3.28; height X = 134.23 cm, SD = 8.25). The study was carried out using the D4S interactive spine and posture rehabilitation system with dynamic, personalised stimulation. Measurements were taken during six successive therapeutic visits of each patient, each containing five repeated sets. A single set involved applying pressure on the device heads 20 times for 8 s. For each patient, the resistance head was in the right top (RT) setting, individually adjusted for patient needs. **Results**: The results showed that the values of the measured moments of the derotating-redressing forces acting on the curvature peak vertebrae ranged from 24.1 N/cm^2^ to 39.9 N/cm^2^. The analysis of the differences in the values of the pressure of the derotating head on the spine in subsequent measurements, accounting for all measurements taken in subsequent cycles of the study, indicates significant variation in pressure values (K-W = 31.0660, *p* = 0.00029 *). Similar dynamics of changes and variation in the results were noted for the values of the pressure applied with the derotating head in subsequent cycles of the study. In this case, the variation in the results was also statistically significant (K-W = 24.4747, *p* = 0.00018 *). **Conclusions**: The values of forces increase slightly with the subsequent series of exercises. The assessment of the value of forces may be an element of optimal and more effective training plans in the therapy discussed.

## 1. Introduction

Scoliosis is a general term that covers a heterogeneous group of conditions involving alterations in the shape and alignment of the spine, chest, and body. Patients with idiopathic scoliosis present a particular diagnostic and therapeutic challenge [1]. One of the most common types of this problem is adolescent idiopathic scoliosis (AIS), which occurs in 1–12% of the population of children and adolescents [2,3]. During puberty, it may progress quickly, which may translate into significant bodily deformation, disturbed function of organs and systems, and impaired body posture stability [1,4,5]. Three-planar changes in the shape and alignment of the spine are diagnosed at every development stage of patients affected with this problem; however, the majority of cases are diagnosed between the ages of 10 and 18 [6,7,8].

The treatment of patients with scoliosis focuses on restoring the highest possible structural and functional fitness of the body. It also includes the impact on the development of the mental sphere, which is particularly significant in puberty. Invasive and often onerous medical procedures trigger various emotions, which require monitoring, analysis, and cooperation between the patient and the therapeutic team. Supporting and activating patients in this respect may have a beneficial effect on the quality of participation, cooperation, and responsibility for the treatment process [9,10].

For this reason, the treatment of individuals with scoliosis should include early and comprehensive activities in the areas of diagnosis, prevention, and therapy, including the psychological aspects, taking into account three-planar changes in spine alignment and the psychological state of the patients.

Physiotherapy of patients with idiopathic scoliosis presents a challenge resulting in a variety of outcomes, not always positive for the patient. The main difficulty is the absence/paucity of clear scientifically proven knowledge concerning the causes of the disorder. Therefore, symptomatic treatment is applied [1]. In scoliosis, it is known that various forces acting on the spine have adverse effects, causing three-planar changes in its alignment. That is why a three-planar therapy taking into account the patient’s current capacity is recommended [1,11]. In the conservative treatment of patients with scoliosis, various therapeutic methods that follow recommendations are known and applied, such as SEAS (Scientific Exercises Approach to Scoliosis) in Italy, BSPTS (Barcelona Scoliosis Physical Therapy School) in Spain, the Side Shift approach in the United Kingdom, global postural re-education (GPR) and the Lyon method in France, DoboMed and FITS in Poland, and the Schroth method in Germany, which is a unique and very popular method, the effectiveness of which has been confirmed by the results of studies [12,13]. Also, special devices for treating patients with scoliosis are described, including the FED method developed by Sastre, using a special device to correct spine alignment in three planes while a computer-controlled pneumatic arm applies the appropriate doses of pressure on the spine. The pressure values are individually adjusted to the patient’s capacity and needs [14,15,16]. As a result of unique exercises applied in the above methods, internal and external forces are generated to correct the pathological alignment of the spine and body parts [16]. In these methods, physiotherapists decide on the frequency and quality of training using their knowledge and experience. The application of conservative methods does not always have positive outcomes, and, therefore, further study in this respect is needed [17].

Another method, developed in the 1970s, is the PRESSIO method, used to support conservative treatment and preparation for surgery. The method is based on the theory proposed by Robert Roaf, emphasising the importance of the rotational component of spine deformations, and on Cotrel’s propositions, taking into account three-planar EDF (elongation, derotation, flexion) action [18,19]. In the PRESSIO method, the elongation component was replaced with pressing (pressio) the structures of the apex of the deformation curvature, i.e., PDF (pressure, derotation, flexion) is proposed instead of EDF. The manner of pressing with pressure heads allows for a transfer of unfavourable loads from articular processes of the concave side of the curvature to vertebral bodies, which opens the possibility of restoring natural spinal curvatures in the period of biological plasticity of the skeletal system. In order to allow the exertion of force from a third point, two other points are stabilised using a shoulder and hip block. Exercises using the Pressio method offer the possibility of three-planar action on two arches of the deformation curvature at the same time by self-dosed and self-controlled correction in the frontal plane, derotation in the transverse plane, and kyphotisation (kyphosis deepening) in the sagittal plane. The therapy is used on an outpatient basis or as home treatment based on recommendations and follow-up visits.

A patient stabilised in a kneeling position, taking into account elongation, kyphotisation, and derotation, applies pressure to the therapeutic heads placed on the apex or apexes of the deformation curvatures. The pressure is applied by the patient using their own muscle strength, which is a safety feature, preventing any damage to soft tissue structures and bones [12,20]. Maximum values of forces used by patients vary between individuals. The frequency and the number of repetitions are determined by the physiotherapist [21,22].

The assumptions of the Pressio method are invariably based on the PDF principle, but the devices used have been modified over the years to adapt to new technological possibilities. From a simple device using a belt, fixed on wall bars (1973), to a device for treatment on an outpatient basis (1974) and its modifications, DIKS (1990) and Delfin (2006) [20], to Dynamic Individual Stimulation and Control for Spine (DISK4SPINE or D4S), which functions as a diagnostic and therapeutic system. The D4S system contains a diagnostic module and two therapeutic modules. The standing module is used in the therapy of body posture, and the kneeling module is for treating scoliosis. The use of various types of sensors to monitor the patient before, during, and after the therapy and a gamification module contribute to the uniqueness of the method and introduce this therapy system to telemedicine. Real-time monitoring of the course of therapy and also pre- and post-therapy measurements may be stored in the DATABASE, offering the possibility of collecting information and converting it into actual patient therapy [23]. The “Empathica E4” band for monitoring life functions records variations in heartbeat, body temperature, and electrodermal activity to complement the diagnostic process in the D4S system, allowing for a comprehensive assessment of the patient’s condition before, during, and after the rehabilitation [9] (Figure 1, Figure 2, Figure 3 and Figure 4).

In the therapy using pressure applied to deformed spine structures, a problem appears with optimal, safe, and, at the same time, effective selection of forces acting opposite to the curvature planes and the time and frequency of their application.

The results of studies have been published concerning the action of distraction forces on the spine during surgical procedures, thanks to which the safe values of applied forces can be determined, which occasionally caused laminar fractures but did not result in torn ligaments or fractures of the epiphyseal plate [24].

A critical assessment of the use of rigid spinal braces, due to their passiveness, stiffness, and lack of control of the applied force, has led to attempts to determine the value of optimal pressure induced in an innovative soft, active orthosis enabling spine mobility with the use of controlled and adjustable corrective forces [25].

The ontogenetic changeability and etiological variability of patients with scoliosis lead to the need for the individual adaptation of the training. On the one hand, the aim is the best possible corrective effect and its permanence; on the other, there is a possibility of permanent damage to the bone or musculoligamentous structures on which pressure is applied [24]. As a result, it is still problematic to accurately assess the force that should be applied to the spine and plan the quantity and quality of patient loads in the therapy. Usually, the process is programmed based on the experience of the therapists. Knowledge in this area and skilful planning of the training, appropriate for the patient’s capacity, may help improve the effectiveness of the therapy.

The study aimed to assess the dynamics of changes in the torques of derotating and redressing forces acting on the apexes of deformation curvature arches during active, kyphosis-inducing exercises in the D4S device.

## 2. Materials and Methods

### 2.1. Study Design

This observational study complied with the Strengthening the Reporting of Observational Studies in Epidemiology (STROBE) statement: guidelines for reporting observational studies [26]. The study was approved by the Bioethics Committee of the Academy of Physical Education in Katowice, in accordance with the Declaration of Helsinki Declaration (No. 3/2019). The patients/participants provided written informed consent to participate in this study.

### 2.2. Setting

The study was conducted in the Orto-Med Scoliosis and Posture Therapy Practice in Bielsko-Biała. The work was carried out from early March 2021 to the end of June 2021 with a one-month interval (April 2021) due to the then-epidemiological situation in Poland and problems with access to the study group.

### 2.3. Participants

The first author (physiotherapist) enrolled 12 girls aged 9 to 10 years (age X = 9.36, SD = 1.52; weight X = 31.34 kg, SD = 3.28; height X = 134.23 cm, SD = 8.25), who were patients of a local private physiotherapy practice. The patients were recruited from among the therapist’s patients. The experiment started with presenting its objective, discussing the measurement equipment, the principles of data confidentiality, and presenting and collecting written consent to children’s participation in the study. The only preparation of the participants for the study involved wearing unrestrictive sports clothing.

The exclusion criteria included:Patient’s age below 9 or above 10 years;No written consent of the legal guardians to take part in the study;Impaired understanding and difficulties in following instructions.

The inclusion criteria were a referral for rehabilitation from a doctor and a diagnosis of scoliosis.

### 2.4. Outcome Measures

The study used the D4S interactive spine and posture rehabilitation system with dynamic, personalised stimulation (Dynamic Individual Stimulation and Control For Spine and Posture Interactive Rehabilitation, Disc4Spine). The basic functions of the D4S system are diagnosing, continual monitoring of the therapy, correcting defective posture in the sagittal plane, functional correction during everyday activities, corrected body posture stabilisation, and patient education. The D4S system allows for a quick and precise diagnosis of a postural defect, determining its degree, and then adjusting the therapy to the patient’s needs and continually monitoring changes. It involves four modules: (1) the physiotherapy module, (2) the actuator module, (3) the gamification module, and (4) the diagnostic and monitoring module. This study used Modules 1, 2, and 4.

The physiotherapy module is a station for advanced exercises based on the Pressio concept, performed in a four-point kneeling position. It allows for conservative treatment and prevention of scoliosis using the active involvement of the patient’s own force. The arches of the deformed curvature are acted upon in three planes simultaneously. The crucial concept here is the priority of the derotation of the vertebrae before posture correction, i.e., taking into account the fact that a change in the position of vertebral bodies in the transverse plane results in a change to the chest shape, leading to deeper deformation. The second principle applied in the PRESSIO method refers to the manner and direction of the derotating and redressing forces, which must be opposite to the direction of the inducing forces. The patient’s position in the cage for exercises in the four-point kneeling position is stabilised with special brackets for the shoulder girdle and pelvis. The physiotherapist responsible for the course of the entire therapeutic process sets the resistance elements on the apexes of the spine curvatures. The heads are set in different configurations depending on the type of deformation curvature. Then, after being appropriately secured in the D4S module, the patient is asked to perform an exercise known as a cat’s stretch. At this time, the resistance elements correct the deformation in three planes (leading to derotation and correction).

The next module in the Disc4Spine system, the actuator module, is equipped with two resistance elements (heads) that primarily work vertically down, causing derotation of vertebral bodies and then, in the phase of the greatest derotation, make a slide to the side, putting the spine in a hypercorrective position. Each of the heads forming a part of the module is equipped with sensors for the constant measurement of the forces generated by the patient while performing the exercises. For the purpose of this study, the MediLogic system was used with a set of sensors recording the signal of the force during the therapy placed on each of the two resistance elements.

The third component used in the presented study is a diagnostic and monitoring module that enables observation of the patient’s physiological functions during the therapeutic exercises. To this end, the Empatica E4 medically certified wristband device was proposed and used for this study. It is worn by the participant on the wrist of the non-dominant hand according to the manufacturer’s recommendations. The device makes it possible to record photoplethysmographic signals, body temperature, and electrodermal activity, which allows for a comprehensive evaluation of the patient’s condition during the rehabilitation. Additionally, the E4 is equipped with a three-axis accelerometer that monitors the movement of the participant [9,12].

Figure 5 presents a comprehensive measurement system with individual modules.

### 2.5. Intervention

The measurements were taken during six successive therapeutic visits of each patient, each containing five repeated sets. A single set involved applying pressure on the heads 20 times for 8 s. For each patient, the resistance head was in the right top (RT) setting, individually adjusted for patient needs.

The measurements were performed at the same time of day during repeated patient visits to the office. Each patient had an individually selected load, but the head settings and loads were constant for the duration of the experiment. (Figure 6 and Figure 7) Before starting the measurement, the patient always performed the same spine stretching exercises (for about 4 min).

### 2.6. Data Measurement

During the study, all sanitary requirements were met in the form of personal protection measures as well as epidemiological interviews that the medical practice carried out each time before the patient started the therapy.

First of all, during the analysis of the forces recorded with the MediLogic sensors, raw values had to be translated into Fi,j force values expressed in force units (N). For this purpose, the relation presented by DeBerardinis [27] was used:$$F_{i,j}=\frac{64}{255}O_{i,j}a,$$
where Oi,j is the raw value registered by a single sensor and ais the area of the sensor of 1.125 cm^2^.

The total value of the recorded force F for a resistance element was expressed by the following formula:$$F=\frac{64}{255}\left(\sum_{i=1}^{n}O_{i,j}\right)a$$
where *n* is the number of sensors in a resistance element.

In the next step, preliminary processing was carried out for the calculated force values. Not all sensors recorded values, as the contact surface of the resistance element with the patient’s back was smaller than the total surface of the sensor system used. For this purpose, zero columns were removed from the result matrix, and those that, during the whole recording, irrespective of the moment in time, showed a constant but false value of force, resulting from, among other things, the sensor fixing system.

Then, the properties of the signal were determined for the right resistance head. For each 8 s exercise, the following were calculated: maximum and mean force value and standard deviation. An example of changes in force value over time has been presented in Figure 8.

### 2.7. Statistical Analysis

The statistical analysis of the results was carried out using the MedCalcver 22.013 statistical package. The Grubbs method (Grubbs double-sided) was used to assess outlier results. The normality of the distribution of the tested variables was analysed using the Shapiro–Wilk method. The results of the descriptive analysis were presented in the form of mean values (X) and their standard deviations (SD). Due to the small number of participants in the study group, the results were completed with the values of their median (M). For the assessment of the variation in results between subsequent measurements and subsequent cycles of the study, the Kruskal–Wallis (K-W) non-parametric test was used.

## 3. Results

The study group consisted of 12 girls aged 9 to 10 years. Table 1 presents the detailed anthropometric data of the sample. Figure 9 shows the qualification stage.

The study results indicate similar values in subsequent cycles of the study and subsequent measurements. The relatively high stability of the results and their low individual variation were noted. No statistically significant differences were observed in the results, either between subsequent measurements taken in individual series of measurements or between subsequent cycles of measurements (Table 2).

It should be noted, however, that there was a noticeable trend in the obtained values of pressure force of the head on the spine to increase in subsequent measurement cycles and subsequent measurements. This trend is presented in Figure 10.

The change in the value of pressure of the head on the spine was particularly visible when it was analysed in reference to the values obtained in subsequent measurements in all participants (Figure 11).

The analysis of the differences in values of the pressure of the derotating head on the spine in subsequent measurements, accounting for all measurements taken in subsequent cycles of the study, indicates significant variation in the values of pressure (K-W = 31.0660, *p* = 0.00029 *).

Similar dynamics of changes and variation in the results were noted for the values of the pressure of the derotating head in subsequent cycles of the study. In this case, the variation in results was also statistically significant (K-W = 24.4747, *p* = 0.00018 *). The results of this analysis are presented in Figure 12.

## 4. Discussion

Treating patients with scoliosis is still a significant medical and social problem. This results from the fact that the causes of the condition are still unknown, which is related to the lack of documented treatment effectiveness. Various therapeutic concepts are based on various approaches to the aetiology of scoliosis. Their effectiveness is still unsatisfactory. It is therefore not surprising that the search for new methods of diagnosing and treating lateral spinal deformity is still ongoing. There is also insufficient information concerning the measurement of external forces acting on the spine and their optimisation in devices used to support the conservative treatment of scoliosis.

The results of the current study showed that the values of measured torques of derotating and redressing forces acting on the apex of the deformation curvature ranged from 24.1 N/cm^2^ to 39.9 N/cm^2^.

These values most probably reflect the capacities of the examined girls determined by their developmental age. These values were similar in subsequent measurements and cycles of exercises. The stability of the results and their low variation were noted. No statistically significant differences were observed in the values of the measured forces, either between subsequent measurements in individual series of measurements or between the subsequent cycles of exercises for individual participants. Further analysis of the results, however, showed a noticeable trend of an increase in the values of the head pressure force on the spine in subsequent measurements and exercise cycles. A progressive change in the value of the head pressure on the spine was particularly noticeable when the results were analysed in reference to the values noted in subsequent measurements in all examined participants in subsequent exercise cycles. The analysis of differences in the values of the pressure of the derotating head on the spine indicates significant variation in those values. Similarly, significant changes and variations in the results were noted for the values of the results of the pressure of the derotating head in subsequent cycles of the study.

The literature notes the positive results of studies using the FED method in treating patients with two-arch, type I and II scoliosis according to the King–Moe classification, with a Cobb angle of 10–60°, in patients aged 11–17 years. The pressure force was adjusted to the patient’s capacity up to a maximum of 100 kg, and the duration of therapy was 30 min. The patients in the study wore Boston braces for 21–22 h a day and underwent other procedures, e.g., electrostimulation of muscles on the convex side of the deformity and hot compresses. The experiment lasted three weeks. The effectiveness of the applied therapy was noted and expressed as significant positive changes in the angle of trunk rotation (ATR) of the thoracic and lumbar spine and the angle of scoliosis [15].

The fundamental difference between the action of external forces employed in the FED method and the PRESSIO method results from a different application of the forces. In the FED method, the force acts indirectly on the spine through the ribs and costovertebral joints, using an external computer-controlled actuator. In the PRESSIO method, however, the torque is generated by the exercising patient performing active kyphosis-deepening exercises of the spine, controlling the size of the load, and the forces act directly on the apical vertebra of the deformation, on the space between the spinous process and the costal process, to be precise. Since the point of force application used in the FED device is on the patient’s ribs, it can be assumed that the value of the force acting directly on the vertebral bodies is lower than measured. The current study was characterised by the patient’s own, individually dosed force, rather than an external force released by the device, and it was a force of lower values than in the FED device.

The literature also contains information concerning the values of forces acting on the spine during surgical treatment of scoliosis using mechanical distractors. Unfortunately, the safety limits in the application of these procedures are not clearly set or known. The applied distraction forces are generally selected based on earlier experiments and general knowledge. In clinical studies, distraction forces of 300–500 N were frequently applied. Occasionally, it resulted in laminar fractures, but no torn ligaments or epiphyseal plate fractures were noted [24]. The results of the measurements of forces in the current study ranged significantly below the values of distraction forces described in the quoted studies. This means that during the exercises on the D4S device, laminar fractures, ligament damage, or epiphyseal damage should not occur.

In the treatment of idiopathic scoliosis, various types of braces and spinal jackets are also used. Both rigid and dynamic spinal jackets are applied. Ali et al. critically assessed the function of rigid corrective jackets due to their passivity, stiffness, and lack of control of the force they apply [25]. The cited authors believe that excessive exertion may contribute to a deteriorated condition of the spine. They proposed an innovative, active soft brace that makes spine mobility possible with the use of controlled corrective forces adjustable by changing the tension using light, low-power actuators. The pressure force is modelled using a model of the contact point of the belt and pulley and is verified by building a non-standard testing station. An actuator module can regulate pressure within the range of 0–6 Kpa, which is comparable to the adjustable pressure in rigid spinal jackets within the range of 0–8 Kpa.

Providing a more effective therapy requires applying various factors necessary for improving patient treatment quality. The main objective of physiotherapy for patients is to improve or restore their functional fitness. This objective may be achieved through obtaining information about and monitoring the affective state of the patients, reflecting their attitude to exertion. The patient and therapist experience various emotions in the therapeutic process and understanding them may build and strengthen their relations and improve the coordination of activities [28]. At the same time, the analysis of the patient’s functioning during the therapy offers the possibility to control and regulate the experienced emotions to the level that will allow the patient’s recovery. The physiotherapist should systematically consider biopsychosocial factors that affect the patient’s momentary and long-term activity. In the D4S system, the “Empatica E4” module is used for this purpose [9].

### Study Limitation

The study presents results of patients of various ages and with various degrees of spine deformation and morphological composition and expands the knowledge concerning the resistance of bone tissue and muscle fibres. The study was of an experimental nature and involved a small group of patients, which should be extended to include a larger number of participants. A wider group of participants should include patients with various degrees and types of deformation. More studies should be carried out to assess the effects of deformation correction through short- and long-term monitoring.

Nevertheless, there is always a question of how to find the balance between effectiveness and patient safety. From an ethical point of view, patient safety is imperative. How should the forces applied to defective alignment of the spine be selected in order to achieve a lasting effect in different patients with different deformations and at different developmental ages? The medical methods referred to above are applied with various therapeutic outcomes and consequences, which are not always positive [24]. There is a need for broader studies in this respect. The results of the study on the assessment of the torques of forces acting on the spine presented here may help plan safe supporting treatment of spine deformation using the PRESSIO method. They indicate the level of autonomically released derotation and redressing forces acting on the apex of deformation curvature, acceptable for the participants. They also point to the possibility of patients adapting to the loads during exercises, which is indicated by the progression in the value of torques of released forces, both gradually during one session of treatment and in successive sessions and may contribute to the planning of assumptions of the training programme.

The therapy using the D4S device may play comprehensive therapeutic functions through individual adaptation, selection, and monitoring of optimum training loads. The therapy using the DISC4SPINE device may also complement the multi-directional treatment of patients with scoliosis and create the possibility of controlling and managing patients’ mental and affective states.

Due to the above, further studies should be carried out on the effects of training in conservative methods used to treat patients with scoliosis that would measure forces and develop optimal individual training plans for individual patients.

Moreover, the role of exercises using the PRESSIO method on the effects of the correction of the deformation degree, in particular derotating changes within the apexes of the deformation curvature (ATR), still needs to be explained.

## 5. Conclusions

The measured values of forces ranged from 24.1 N/cm^2^ to 39.9 N/cm^2^ and did not differ significantly between individual patients.

The values of the forces increase slightly with subsequent series of the exercises.

## 6. Practical Implications

The assessment of the value of forces may be an element of optimal and more effective training plans in scoliosis therapy.

## Figures and Tables

**Figure 1 jcm-13-07746-f001:**
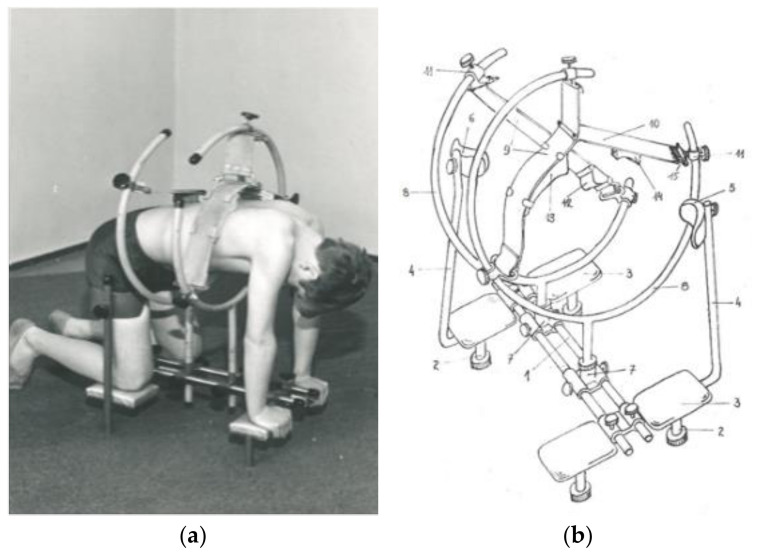
(**a**) Self-corrector, (**b**) construction diagram [22].

**Figure 2 jcm-13-07746-f002:**
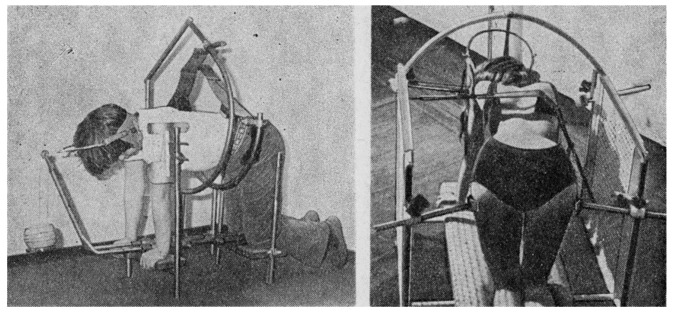
(**left**) A self-corrector with a neck pulley, (**right**) a redressing device used in rehabilitation rooms [21].

**Figure 3 jcm-13-07746-f003:**
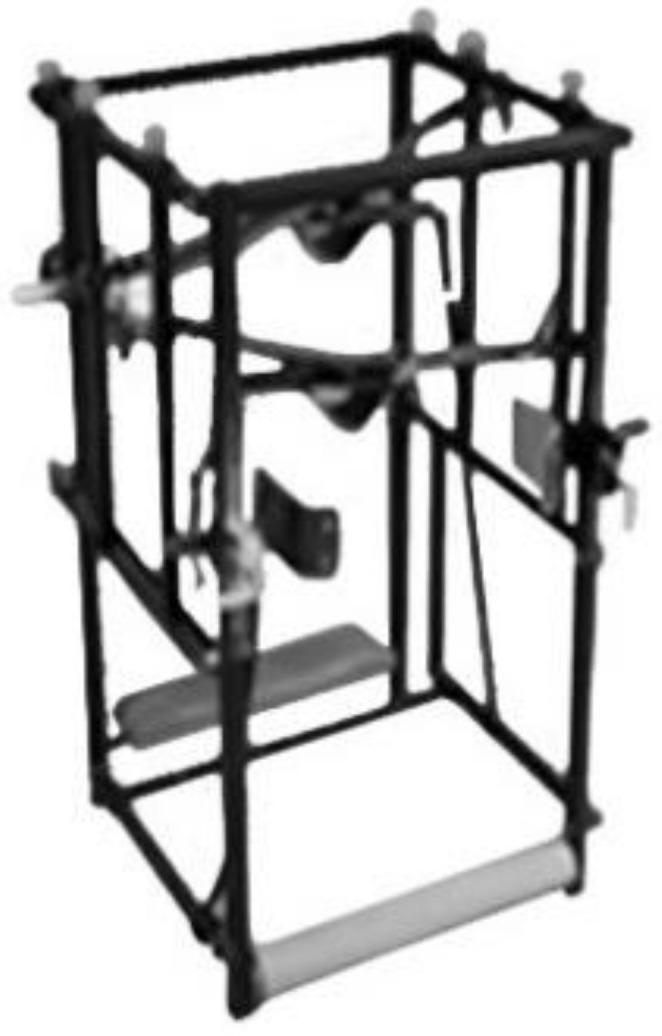
Derotator and scoliosis corrector—DIKS [20].

**Figure 4 jcm-13-07746-f004:**
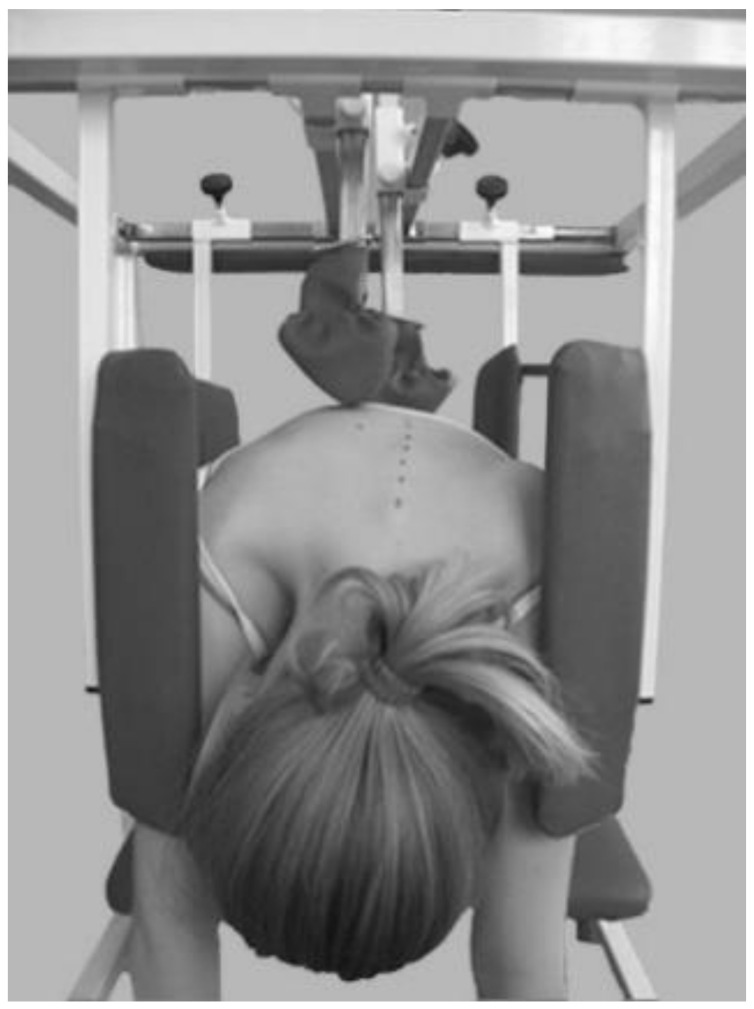
Delfin device [20].

**Figure 5 jcm-13-07746-f005:**
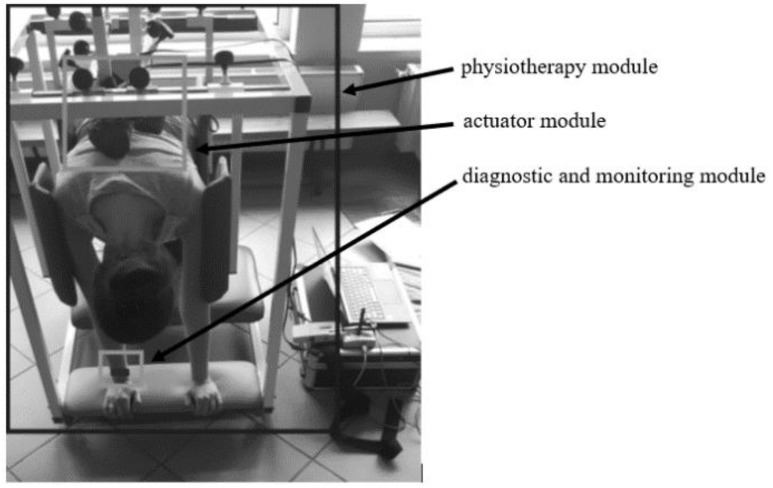
D4S measurement model.

**Figure 6 jcm-13-07746-f006:**
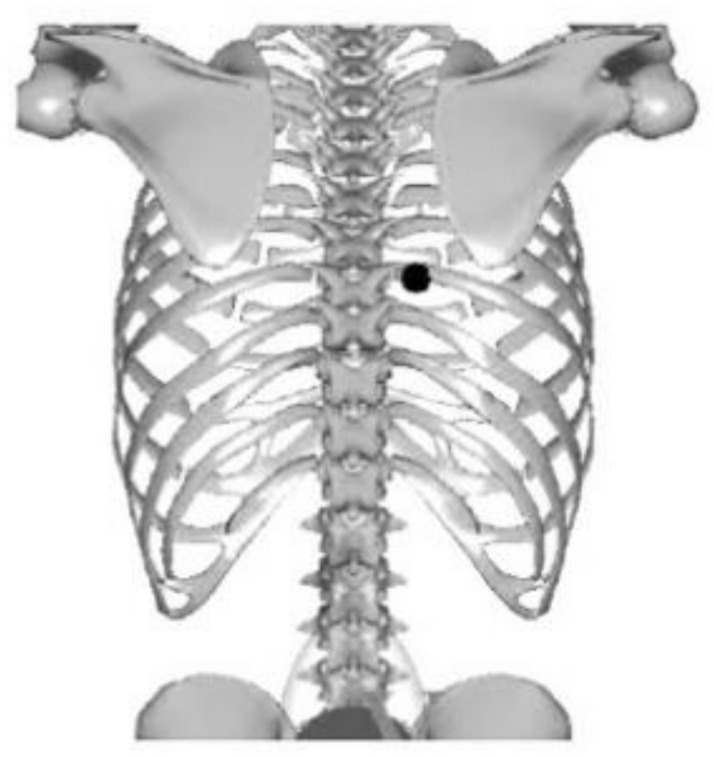
Placement of the head on the patient’s body, overall view.

**Figure 7 jcm-13-07746-f007:**
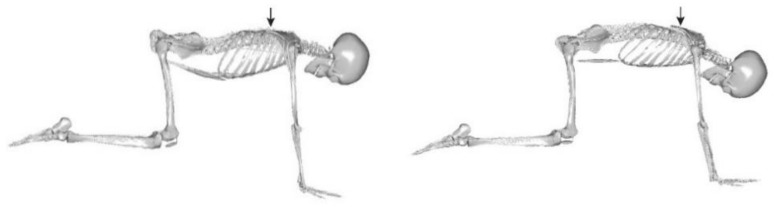
Placement of the heads on the patient’s body, side view.

**Figure 8 jcm-13-07746-f008:**
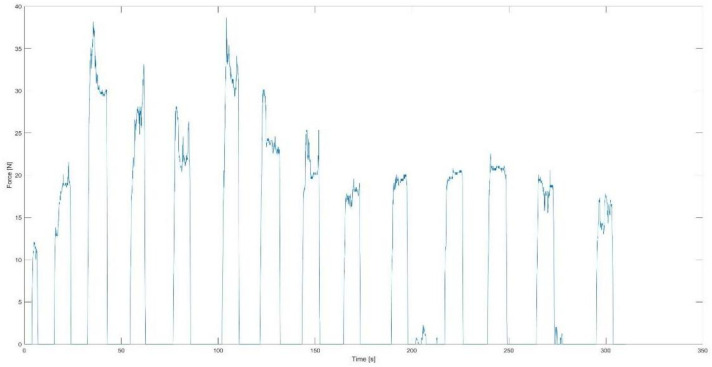
Example of changes in force value over time.

**Figure 9 jcm-13-07746-f009:**
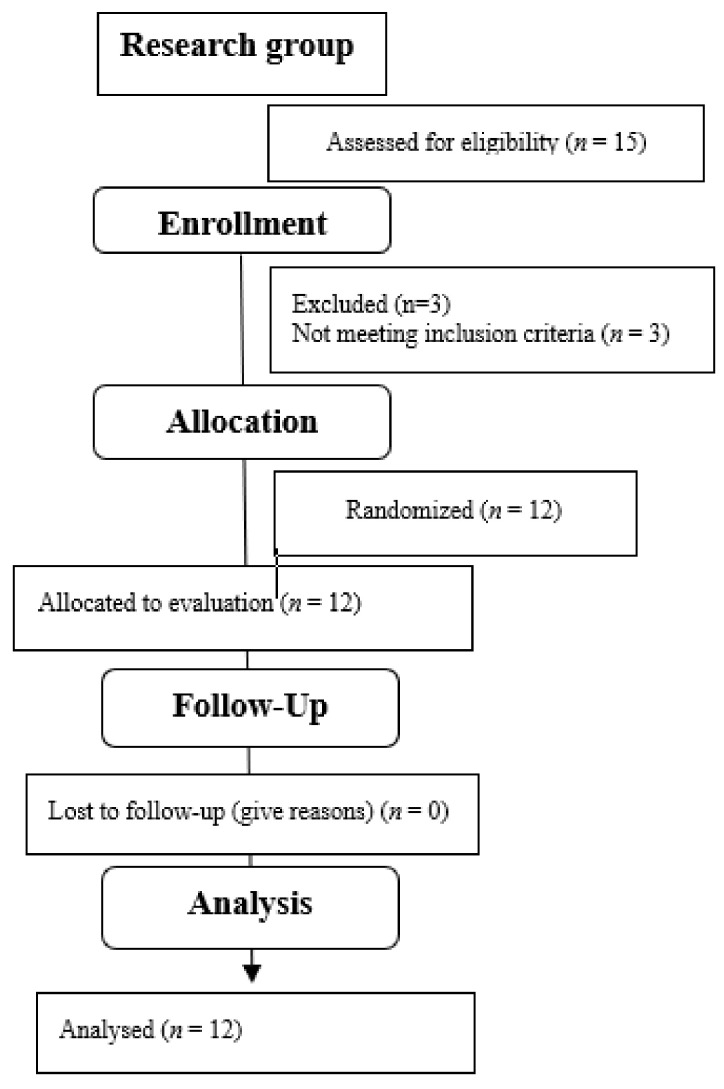
CONSORT Flow Diagram.

**Figure 10 jcm-13-07746-f010:**
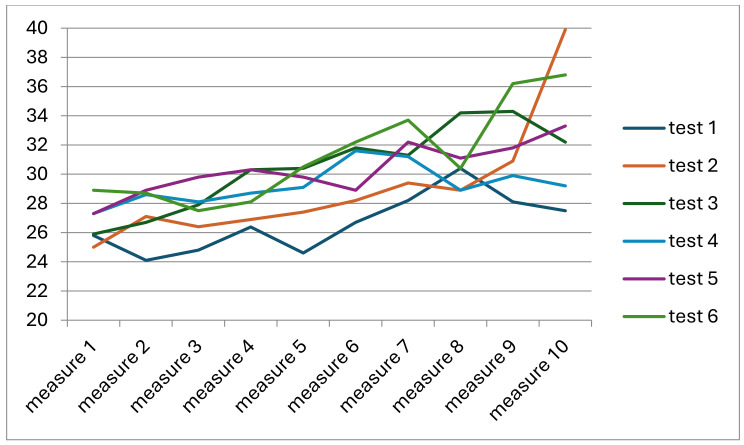
Changes in the value of the pressure force of the derotating head on the spine in subsequent measurements and subsequent measurement cycles.

**Figure 11 jcm-13-07746-f011:**
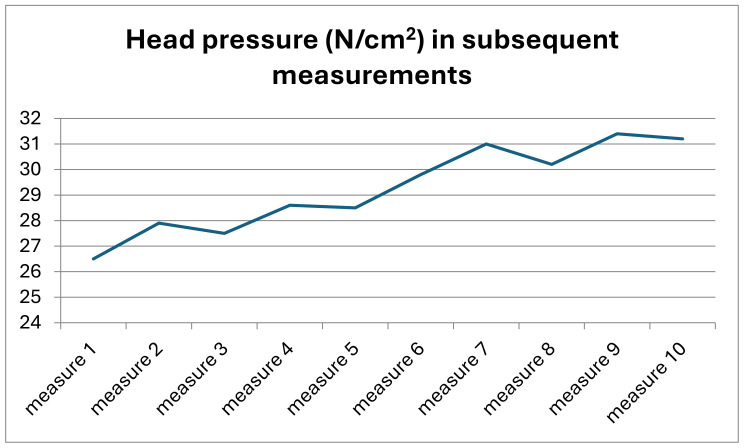
Changes in the pressure of the head on the spine in subsequent measurements in the entire study group.

**Figure 12 jcm-13-07746-f012:**
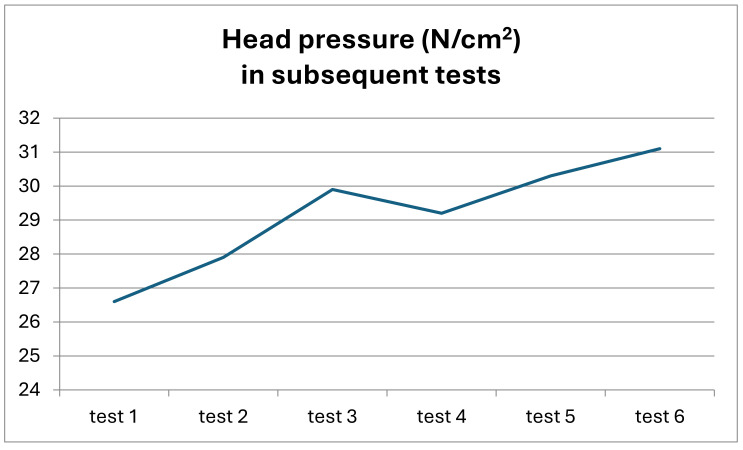
Changes in the pressure of the head on the spine in subsequent tests of the study in the entire study group.

**Table 1 jcm-13-07746-t001:** Anthropometric data.

Variable	Research Group *n* = 12
	X ± Std
Weight (kg)	31.34 ± 3.28
Height (cm)	134.23 ± 8.25

*n*—number of participants; X ± Std—mean ± standard deviation.

**Table 2 jcm-13-07746-t002:** Values of head pressure on the spine in subsequent measurements.

Head Pressure on the Spine (N/cm^2^) *n* = 12							
Test Measurement	Test 1	Test 2	Test 3	Test 4	Test 5	Test 6	K-W Test *p*
	X ± SD Median	X ± SD Median	X ± SD Median	X ± SD Median	X ± SD Median	X ± SD Median	
measurement 1	25.8 ± 5.47 25.3	25.0 ± 9.01 27.5	25.9 ± 6.34 26.8	27.3 ± 6.09 25.1	27.3 ± 6.03 27.3	28.2 ± 9.44 28.1	*p* = 0.9721
measurement 2	24.1 ± 7.03 24.3	27.1 ± 8.02 28.4	26.7 ± 6.23 24.1	28.6 ± 5.81 29.4	28.9 ± 6.51 28.9	28.7 ± 9.24 29.0	*p* = 0.6738
measurement 3	24.8 ± 6.49 22.9	26.4 ± 7.83 26.9	27.9 ± 6.53 27.5	28.1 ± 5.76 28.4	29.8 ± 6.35 28.8	27.5 ± 10.86 29.0	*p* = 0.6327
measurement 4	26.8 ± 8.26 25.8	26.9 ± 8.32 28.4	30.3 ± 6.83 31.5	28.7 ± 7.17 28.7	30.3 ± 6.29 29.1	28.1 ± 7.28 28.3	*p* = 0.7562
measurement 5	24.6 ± 6.07 24.9	27.4 ± 7.62 27.3	30.4 ± 8.47 32.9	29.1 ± 7.09 29.1	29.8 ± 6.98 29.8	30.5 ± 7.61 30.3	*p* = 0.5025
measurement 6	26.7 ± 7.08 28.2	28.2 ± 8.22 28.5	31.8 ± 7.95 33.7	31.6 ± 6.62 31.9	28.5 ± 9.19 29.8	32.2 ± 8.24 31.9	*p* = 0.4845
measurement 7	28.2 ± 7.06 27.	29.4 ± 7.22 28.6	31.3 ± 7.72 34.2	31.2 ± 6.22 30.8	32.2 ± 7.01 31.6	33.7 ± 9.05 31.0	*p* = 0.8036
measurement 8	30.4 ± 10.55 29.7	28.9 ± 5.94 27.9	34.2 ± 6.71 34.2	28.9 ± 7.85 25.5	31.1 ± 6.18 30.7	30.4 ± 6.34 31.6	*p* = 0.8954
measurement 9	28.1 ± 4.23 27.9	30.9 ± 4.89 32.6	34.3 ± 7.72 34.3	29.9 ± 9.26 34.9	31.8 ± 5.72 33.4	36.2 ± 9.09 38.2	*p* = 0.1805
measurement 10	27.5 ± 5.75 26.3	39.9 ± 7.40 29.6	32.2 ± 6.51 32.2	29.2 ± 6.46 28.9	33.3 ± 6.52 32.6	36.8 ± 7.89 33.0	*p* = 0.1082
K-W Test *p*	*p* = 0.7151	*p* = 0.7666	*p* = 0.3063	*p* = 0.9085	*p* = 0.5000	*p* = 0.5188	

*n*—number of participants; X ± Std—mean ± standard deviation; K-W Test—Kruskal–Wallis test; *p*—*p*-value.

## Data Availability

The minimal data set is contained within our paper.

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
