# Peer review of "The Assessment of the Dynamics of Changes in the Torques of Redressing and Derotating Forces Acting on the Spine During Active Kyphosis-Deepening Exercises in the Dynamic Individual Stimulation and Control for Spine Device (DISC4SPINE, D4S)"

_jcm, 2024, doi:10.3390/jcm13247746_

Round 1

Reviewer 1 Report

Comments and Suggestions for Authors

The authors introduced very important idea about: The assessment of the dynamics of changes in torques of  redressing and derotating forces acting on the spine during  active kyphosis-deepening exercises in the Dynamic Individual Stimulation and Control For Spine device (DISC4SPINE, D4S).

-The introduction

Too long, you need to reduce it, and figure 2 is not obvious.

-Materials &Methods

Explain lines 189-200. Not clear.

Please give more details about the intervention.

Give references for D4s System.

Figure 5 not obvious.

Comments on the Quality of English Language

The English could be improved to more clearly express the research.

Author Response

Thank you very much for your time and experience in providing feedback on the manuscript as well as valuable comments and suggestions regarding the article. Below is the answer to your review.

Reviewer #1: 
The authors introduced very important idea about: The assessment of the dynamics of changes in torques of  redressing and derotating forces acting on the spine during  active kyphosis-deepening exercises in the Dynamic Individual Stimulation and Control For Spine device (DISC4SPINE, D4S).
1.    The introduction
Too long, you need to reduce it, and figure 2 is not obvious.
Response: The introduction aimed to provide a historical outline of the development of the Pressio method. This allows for understanding the idea behind the D4S methodMaterials &Methods
•    Explain lines 189-200. Not clear.
Response: Outcome measures in lines 189-200 are an introduction to the description of the research tool, what it can be used for. Due to the editorial limitations of the journal, it has been presented in a short form. A more detailed description of the tool is given in the references:
1.    Romaniszyn-Kania, P.; Pollak, A.; Danch-Wierzchowska, M, et al. Hybrid System of Emotion Evaluation in Physiotherapeutic Procedures. Sensors 2020, 20, 6343. doi: 10.3390/s20216343
2.    Szurmik, T.; Bibrowicz, K.; Lipowicz, A.; Mitas, AW. Methods of Therapy of Scoliosis and Technical Functionalities of DISC4SPINE (D4S) Diagnostic and Therapeutic System. In: Pietka E, Badura P, Kawa J, Wieclawek W. (eds) Information Technology in Biomedicine. Advances in Intelligent
•    Please give more details about the intervention.
Response: the description of the intervention has been expanded
„The measurements were performed at the same time of day during repeated patient visits to the office. Each patient had an individually selected load, but the head settings and loads were constant for the duration of the experiment. Before starting the measurement, the patient always performed the same spine stretching exercises (for about 4 minutes).”
•    Give references for D4s System.
Response: references for the D4S method have been added
•    Figure 5 not obvious. 
Response: Figure 5. is an example of a real measuring station with additional modules that was used in the research.

Reviewer 2 Report

Comments and Suggestions for Authors

Authors present a study on 12 female patients aged 9-10 years with adolescent idiopathic scoliosis AIS to assess the dynamics of changes in torques of  redressing and derotating forces acting on the spine during  active kyphosis-deepening exercises in the Dynamic Individual  Stimulation and Control For Spine device (DISC4SPINE, D4S). The values of forces has been shown to increase slightly with subsequent series of exercises. Introduction provides a lot of information on different forms of physical therapy. Although number of patients is low, Materials and Methods are clearly defined and the study is well organized from that aspect. Discussion is extensive and covers all aspects of the paper, however one thing is not very clearly defined - clinical applications of these findings and potential future advances - what is the consequence of the fact that the value of applied forces increase?

Author Response

Thank you very much for your time and experience in providing feedback on the manuscript as well as valuable comments and suggestions regarding the article. Below is the answer to your review.

Reviewer #2: 
Authors present a study on 12 female patients aged 9-10 years with adolescent idiopathic scoliosis AIS to assess the dynamics of changes in torques of  redressing and derotating forces acting on the spine during  active kyphosis-deepening exercises in the Dynamic Individual  Stimulation and Control For Spine device (DISC4SPINE, D4S). The values of forces has been shown to increase slightly with subsequent series of exercises. Introduction provides a lot of information on different forms of physical therapy. Although number of patients is low, Materials and Methods are clearly defined and the study is well organized from that aspect. Discussion is extensive and covers all aspects of the paper, however one thing is not very clearly defined - clinical applications of these findings and potential future advances - what is the consequence of the fact that the value of applied forces increase?
Response: the results obtained indicate that patients adapt to the effort and this is an observation that allows for more effective planning of training/therapy for patients with scoliosis. It also allows for individualized therapy, which in turn will avoid overloading the musculoskeletal system

Reviewer 3 Report

Comments and Suggestions for Authors

The article is compelling and well organized. The abstract clearly summarizes the study objectives, methodology, results, and their significance. The aim of the study is to assess the dynamic changes in the torques of derotating and redressing forces acting on the apexes of deformed curvature arches during active kyphosis-inducing exercises using this D4S device. The results indicated that the measured moments of the derotating and redressing forces acting on the apex vertebrae ranged from 24.1 N/cm² to 39.9 N/cm², highlighting the effectiveness of the treatment approach.

In the introduction, the authors provide a clear overview of the importance of addressing three-planar changes in the shape and alignment of the spine in patients with scoliosis. 

The materials and methods section is complete and well explains the processes undertaken to measure the forces and torques in the D4S device. The methodology is described in sufficient detail, allowing for reproducibility and understanding of the experimental design.

The results are presented clearly, showcasing the statistical analysis and the range of values observed. Tables and figures effectively illustrate the key findings, making the data accessible for readers.

The discussion effectively connects the study's findings with existing literature, offering a comprehensive analysis of the implications of these results on the treatment of scoliosis. The authors relate their findings to previous studies, reinforcing the significance of early intervention and combined therapeutic approaches.

One detail: the unit cm² should be noted with the "2" written in superscript (cm²) to maintain scientific accuracy.

Author Response

Thank you very much for your time and experience in providing feedback on the manuscript as well as valuable comments and suggestions regarding the article. Below is the answer to your review.

Reviewer #3: 
One detail: the unit cm² should be noted with the "2" written in superscript (cm²) to maintain scientific accuracy.
Response: the unit notation per cm2 has been corrected

Reviewer 4 Report

Comments and Suggestions for Authors

The study is interesting, however, I have some concerns to be discuss.

  1. What was the primary objective of the study involving the D4S device?
  2. How were the participants of the study characterized in terms of age, weight, and height?
  3. What were the key findings related to the range of values of the measured moments of the derotating-redressing forces?
  4. What statistical significance was found in the variation of the derotating head pressure values over subsequent cycles of the study?

Author Response

Thank you very much for your time and experience in providing feedback on the manuscript as well as valuable comments and suggestions regarding the article. Below is the answer to your review.

Reviewer #4: 
1.    What was the primary objective of the study involving the D4S device?
Response: in lines 157-159 it is given: The aim of the study was to assess the dynamics of changes in torques of derotating and redressing forces acting on the apexes of deformation curvature arches during active, kyphosis inducing exercises in the D4S device.
2.    How were the participants of the study characterized in terms of age, weight, and height?
Response: Table 1. presents the anthropometric data of the patients. A scale and a height gauge were used for this purpose. Additionally, a Flow Diagram is provided, which characterizes the course of patient qualification
3.    What were the key findings related to the range of values of the measured moments of the derotating-redressing forces?
Response: Own research did not show any unequivocally statistically significant differences in the Kruskal Willis test, however, observation of the results indicates a clear upward trend in the values of forces in subsequent measurements and measurement cycles. The forces increased in subsequent measurements without statistical significance, but a clear upward trend in the values of derotational and retractive forces was observed in all the subjects, which may indicate the body's adaptation to effort and loads. This fact may be helpful in planning future training using D4S.
4.    What statistical significance was found in the variation of the derotating head pressure values over subsequent cycles of the study? 
Response: No significant differences in the values of the head derotation forces were found, however, an upward trend was observed in all subjects in subsequent measurements and cycles.

Round 2

Reviewer 1 Report

Comments and Suggestions for Authors

No comments.

Reviewer 4 Report

Comments and Suggestions for Authors

The manuscript is suhtable for publication.